# Real Estate Investment, Loan Preference, and National Happiness: Evidence from China

**Shangan Ke** [1],* and **Xinhai Lu** [2]

1   School of Public Administration, Central China Normal University, Wuhan 430079, China
2   College of Public Administration, Huazhong University of Science and Technology, Wuhan 430074, China; xinhaiu@mail.ccnu.edu.cn
*   Correspondence: keshangan@ccnu.edu.cn

**Abstract:** With two-stage least squares and LS models, this paper investigates the effects of real estate investment and loan preference on national happiness with data of 31 provinces in China from 2000 to 2018. The conclusions are that a rapid growth of real estate investment has significant adverse effects on national happiness in modern China. Specifically, real estate investment has negative impacts on disposable income, consumption structure, and personal development. Moreover, the rapid growth of real estate investment and institutions' loan preference leads to other inhibitory effects on national happiness. The intermediary effect model confirms the transmission mechanism of the rapid growth of real estate investment, the loan structure, and national happiness. According to our research, we propose conductive suggestions for the governments.

**Keywords:** real estate investment; national happiness; loan structure; inhibitory effect; China

## 1. Introduction

National happiness refers to a series of joyful and pleasant emotions that human beings subjectively produce based on their sense of satisfaction and security, affected by the external economy, social environment, and psychological state. However, the World Happiness Database indicates that people do not feel happier as expected with economic growth. Since the Reform and Opening, the world has witnessed the world-shaking achievements of China in economic development. Unfortunately, China also fell into a Happiness Stagnation predicament at the same time. The World Happiness Database shows that China's average national happiness fell from 7.3 in 1990 to 6.5 in 2000, to 5.0 in 2012.

As the engine of economic development, the rapid expansion of real estate investment and the continual rise in housing prices may negatively affect national happiness [1]. Infact, in recent years, the literature has been concerned about such issues. Wang, Hou, and He studied the dynamic linkage between housing prices and enterprises' investment behaviors in China [2]. They found that the rapid uprising housing prices significantly affected the enterprises' investment behaviors, expanding the gap of financial constraints between private and state-owned enterprises, leading to investment inefficiency and resource mismatch [1].

Even though the governments made and executed the policies of restraining demands and adding land supply to ease the real estate bubble, the net impacts of the rapid rise of real estate investment on other industrial factors are still negative. What matters is that scholars found that the national happiness level is tightly bonded to their housing level [3]. Li et al. also found that there were significant differences in the impacts of self-owned housing levels. Larger housing can significantly improve householder's happiness, and the marginal effect of housing units on national happiness is diminishing [4]. Further research found that increasing the sales area of ordinary residential and affordable housing will enhance national happiness, while villa and high-end apartment demands play the

opposite role. The higher the price of real estate, the more challenging it to raise the happiness index [5].

In this scenario, what attracts us is whether the rapid growth of real estate investment affects national happiness, influencing society's sustainable development and the economy. These issues have become one of the hot concerns for scholars and policymakers. Objectively speaking, if neglecting the negative impacts, it is impossible to understand the external constraints in improving national happiness. This research highlights China's rapid real estate investment, especially its negative effect on national happiness, with panel data of China's provincial-level from 2000 to 2018. Considering the special relationship between real estate and financial institutions, we need to explore the inhibitory effect of financial institutions' behaviors on national happiness, which prefers to meet the long-term loan demand of real estate sectors.

The main contributions of this paper are as follows. Although existing research has focused on the relationship between housing and residents' happiness, more on the social and economic value of housing itself, few scholars have paid attention to the impact of continued growth in real estate investment on residents' happiness. In China's economic and cultural environment, large-scale real estate investment and speculation have caused housing prices to remain high, and the symbolic significance of wealth and status attached to real estate has been highly magnified. Its impact on Chinese residents' happiness is undeniable. Studying the mechanism of China's real estate investment's influence on residents' happiness can enhance theoretical research on real estate investment and happiness. It can also help the government better cope with real estate investment's negative impact on residents' happiness.

The existing literature does not pay attention to the critical influence of loan preference on the relationship between real estate investment and residents' happiness. This research can help us to understand how loan preference affects the relationship between the two factors and deepen the understanding of loan preference's influence on the "real estate investment–resident happiness" framework. The intermediary effect of "sense" reveals the mechanism and influence of intermediary effect, and at the same time, can provide policy reference for decision-making departments.

## 2. Literature Review and Theoretical Assumptions

### 2.1. The Influence Factors of National Happiness

The concept of national happiness involves subjective feelings and objective environments and is closely related to personal abilities, social environment, and inner feelings. The psychological state is affected by the external environment, and the psychological state has an individual effect. The subjectivity is extreme, and it is impossible to measure the nation's overall happiness level from the macro level. Therefore, this research will adopt the social and economic environment of happiness as the proxy variable of national happiness. According to the existing literature, we select the indicators that affect national happiness and summarize the main influencing factors.

Specifically, (a) Income growth, represented by the per capita disposable income growth rate, marked with *income_increase*. Although national happiness decreases rapidly with the increase in income, the positive relationship between income and happiness has been proven by many scholars [6–8]. (b) Average disposable income. Transnational evidence shows that the relationship between economic development and happiness is of curve-type. The per capita income of USD 10,000 is the critical point. Below this critical point, as the per capita income increases, happiness is also greatly improved. Therefore, per capita, disposable income should be included in the agent indicators of national happiness at China's current income situation with the identification code *income.* (c) Gini coefficient. National happiness will adjust to income with the double effects of relative income and total income. Graham, Carol, and Stefano analyzed 17 Latin American countries and Russia's happiness and found that the relative income gap impacted how individuals rated their happiness [9]. Alesina and Perotti verified the conclusion in 1996 [10]. Therefore,

another agent indicator of national happiness is designed as income inequity, represented as the Gini coefficient, identified with *gini_coefficient*.

(d) Inflation and unemployment. Despite the criticism, there is still evidence that unemployment and inflation do reduce national happiness, which coincides with the welfare theory proposed by Andrew Clark and Andrew Oswald. Inflation weakens purchasing power and makes people feel that their morale and national prestige are weakened and feel exploited. In addition to economic losses, unemployment also leads to loss of self-esteem, depression, anxiety, and society's humiliation. Blanchflower and Oswald analyzed data from Europe and the United States; they found that happiness, inflation, and unemployment have a strong correlation [11]. Thus, the other two surrogate indicators of happiness are set to the unemployment rate (expressed as the registered unemployment rate) and the inflation rate (expressed in CPI growth rate), with the identification codes *unployment_rate* and *inflation_rate*.

### 2.2. Influence and Mechanisms of Real Estate Investment on National Happiness

China is rising, where national life quality and consumption level, to a degree, determine national happiness sense. Scholars have found the significant inhibitory effects of real estate investment—the first cost effect. The growth of real estate investment results in high housing prices, which leads to the inhibitory effect on consumption demand. In China, people increase their savings rate by decreasing the current consumption demand to meet future housing demand. Wang and Wen analyzed how real estate investment leads to a nation's resource mismatch. They found that the rapid growth of real estate investment through interest rate inhibits individuals' consumption [12]. It is easy to find the investment in real estate sectors increased from 7.8% in 1996 to 14.4% in 2007, and the share of real estate sales in GDP increased fast after 2004 [5]. More than one-quarter of income was spent on housing in 2006, and it was more than 40% in Chinese cities (the data were provided by China's Economic and Social Development Statistics Database). The increase in living costs means the sacrifice of other demands. The high housing prices have caused negative impacts on national happiness through consumption mismatch and work selection [13].

Secondly, is the symbolic effect. Real estate property has symbolic meanings of wealth and social status beyond living function in China. Since the restructuring of the real estate sector in 1997, real estate speculation has led to a rapid rise in housing prices. Housing is a place to live and investment goods with tremendous value, a symbol of social status and wealth. The Youth Development Research Center survey showed that housing ownership has significant impacts on dwellers' socio-economic status [14]. Housing has become an essential factor affecting youth's urban life, mainly for marriage, birth, and education. As a bond between the market and the government, the real estate sector plays a more critical role in strengthening the middle-class structure and is an essential indicator of social–economic stratification [15]. Due to the symbolic effect of housing in China, low-income groups, especially young people, face tremendous economic pressure and social pressure, and happiness is significantly inhibited. Based on the symbolic effect, the literature has also proved that sometimes real estate investments may lead not to happiness, but stigma and unhappiness. For example, a state investing estate company in Hungary built new residences for marginal people, leading to over-marginalization and unhappiness [16].

Thirdly, there is the siphoning effect. The large income gap between the real estate sector and others will exacerbate social injustice and cause national happiness loss. The high profits of the real estate sector will have a siphon effect through the market's configuration of the labor force, restraining other sectors' development. Before 2014, the real estate industry's average net profit was more than 30%, while the Chinese industrial enterprises were not more than 7%. Moreover, top first 500 industrial enterprises' average net profit was 2.3% [17]. The considerable gap in profits leads to a mass transfer of resources.

Moreover, the income gap between real estate and non-real estate sectors easily sharpens social contradictions and inequity. According to the analysis above, we believe

that the rapid growth of real estate investment affects national happiness negatively, in theory. To verify the above analysis, we propose the following hypothesis:

**Hypothesis 1 (H1):** *in the period 2000–2018, the rapid growth of real estate investment had a significant hindrance on national happiness.*

### 2.3. The Distorting Effects of Loan Preference

Loan preference refers to the preferred choice of lenders of financial institutions embodied in the loan structure. The profit in real estate sectors is higher than most sectors in contemporary China, and banks tend to lend to avoid risks, which motivates banks to increase the loans for low-risk, high-yielding real estate sectors and decrease the loans for individuals.

Huang et al. also found that the adjusted capital adequacy ratio will affect the risk appetite of banks' credit and behaviors under the new Basel Agreement [18]. The agreement causes a bank credit crunch and eventually motivates banks to reduce the loans to higher risk and lower profit units, such as small and medium enterprises, and individuals. Banks prefer lower risk and higher profit units, such as real estate sectors [12]. Mohammad et al. studied the American real estate credit market and found that real estate prices were near bank loans. Banks prefer to offer loans to real estate developers and homebuyers [19]. However, bad real estate loans will increase the lending default rates, bank credit risk increases, and the real estate bubble could, which could even cause financial crises to develop [18]. The rapid growth of real estate investment on the "Crowding-in effect of personal credit" and the rise of economic risks caused by bad real estate loans will eventually lead to economic malformation [2,20,21].

According to the data provided by the China Real Estate Financial Report over the years, the proportion of China's real estate development funds from bank financing from 1993 to 2013 exceeded 40%, and even up to 50% to 60%. Real estate loans accounted for the proportion of long-term loans to financial institutions which have gradually increased. Zhang et al. found that at the end of 2002, this ratio reached 27.5%, and in 2005 it increased to 34.8% [17]. It further confirms that China's banking system will prioritize the real estate sector investment needs of the basic facts under the financial institutions' leadership. Based on the above analysis, we believe that the rapid growth of investment in the real estate sector may be through the banking-based financial institution distorting effect, which exacerbates the residents' sense of happiness. Therefore, the second hypothesis is as follows:

**Hypothesis 2 (H2):** *from 2000 to 2018, the growth rate of real estate investment significantly inhibited national happiness, and loan preference enlarged the inhibiting effect.*

### 3. Empirical Model and Identification Strategies

#### 3.1. Econometrical Model Design and Data Sources

According to the hypotheses above, the empirical model is to verify whether real estate investment growth significantly impacts national happiness. The model is set as follows:

$$happiness_{it} = \beta_0 + \beta_1 realestate\_increase_{it} + \delta Z + \varepsilon_{it} \tag{1}$$

In Equation (1), the variable $happiness_{it}$ represents the level of national happiness at $t$ year of $i$ city, $realestate\_increase_{it}$ is the indicator of real estate investment growth at $t$ year of $i$ city, Z is a variable set, and $\varepsilon_{it}$ is a random disturbance term. According to the literature review in Section 2.1., we use income growth, average disposable income, Gini coefficient, inflation rate, and the unemployment rate as the agent indicators of national happiness.

To minimize the bias of the results caused by the missing variables in Equation (1), we set the variables in the control variable set *Z*, including (a) real average GDP growth rate (*pergdp_growth*). The data were from the China Statistical Yearbook and the China

Statistical Yearbook. The average real GDP was calculated with total real GDP divided by the total population of *t* year and *i* province. GDP per capita controls the impact of economic development on national happiness.

(b) Educational capital (*educationcapital*). Educational capital is an essential factor affecting national happiness [22,23], measured with a reciprocal ratio of illiterate population aged 15 and above in an area.

(c) Health capital (*healthcapital*). Good health status is the prerequisite of residents earning income to improve life quality and the vital part of human capital [24]. Given the close relationship between health and happiness, it is essential to control the health capital variable. Health status is measured with the reciprocal of total population mortality rate at regions.

(d) Industrial structure (*industry_structure*). Robinson, Chanel, and Fader start with the basic neoclassical growth model and join structural variables to study economic growth [25,26]. The results show that structural factors have a significant contribution to economic growth. Additionally, the industrial structure affects national happiness through residents' income.

(e) Tax rate (*tax*). Tax level and tax policy significantly impact happiness based on behavioral economics [26]. Tax can achieve social distribution justice and improve national happiness by "rob the rich to assist the poor" during social redistribution [27].

(f) Social security (*social_security*). Social security is an essential method of income redistribution. The multi-level and multi-dimensional guarantee of primary living can significantly affect people's happiness, especially those with low incomes. The research indicates U-type relationships between other social security levels and national happiness [28].

Moreover, the research controls the dummy variables of fixed effects in provinces and years in Equation (1) for differences between different provinces and years in macro-environment and policy orientation. Considering that the influence of certain control variables on happiness in Equation (1) has a lagged effect, the control variables are lagged by one period.

### 3.2. Discussion about Stochastic Explanatory Variables and Instrument Variable Design

From the econometric model's logical analysis, there may be random explanatory variables between the growth rate of real estate investment and national happiness in China. First, there are two reasons: the two-way causal relationship between the growth rate of real estate investment and national happiness. Secondly, the missing variables issue is challenging to avoid. The introduction of instrumental variables is a feasible solution to this problem. According to the instrument variables' introduction rules, when the number of the introduced instrument variables is greater than or equal to the number of endogenous variables, the instrumental variable method is proper.

We introduce an instrument variable equal to the number of endogenous variables (real estate investment growth rate), so the instrument variable of land supply can be identified. Based on the analysis above, this research uses the transferred area per capita of construction land (the transferred area of construction land in each area / the population of each area) lagged one period as the instrumental variable, with the identified code *perconstruction_land*. The reasons are as followed. First, calculating the transfer area per capita for construction land can eliminate the construction land area differences in regions. Second, the lagged phase can eliminate the two-way causal relationship between real estate investment and construction land. Therefore, the measurements of instrument variables are:

$$happiness_{it} = \beta_0 + \beta_1 realestate\_increase_{it} + \delta Z + \varepsilon_{it} + \beta_2 W_{it} \qquad (2)$$

$$realestate\_increase_{it} = \pi_0 + \pi_1 perconstruction\_land_{it} + \pi_n W_{it} + v \qquad (3)$$

In Equation (2), $W_{it}$ is an exogenous variable not related to the error term. In Equation (3), $\pi_0$, $\pi_1$, $\pi_n$ are coefficients, and $v$ is the error term.

This research adopts the entropy weight coefficient method to assign a weight to the five indicators of national happiness and work out each province's national happiness score. The main steps are as follows.

There are differences in the dimensions and magnitude of national happiness indicators and the indicators' positive and negative signs, and the indicator data need to be standardized. We standardize the positive and negative indicators separately.

① Data dimensionless positive index:

$$X'_{ij} = (X_{ij} - \min X_j) - (\max X_j - \min X_j) \tag{4}$$

Negative indicators:

$$X'_{ij} = (\max X_j - X_j) - (\max X_j - \min X_j) \tag{5}$$

② Calculate the proportion of $X_{ij}$:

$$Y_{ij} = X'_{ij} / \sum_{i=1}^{m} X'_{ij} \tag{6}$$

③ Calculation of index information entropy:

$$e_j = -k \sum_{i=1}^{m} (Y_{ij} \times LnY_{ij}) \tag{7}$$

If $k = \frac{1}{Lnm}$, then $0 \leq e_j \leq 1$, and when $Y_{ij} = 0$, $Y_{ij} \times LnY_{ij} = 0$;

④ Information redundancy calculation:

$$d_j = 1 - e_j \tag{8}$$

⑤ Calculation of indicator weight:

$$w_i = d_j / \sum_{j=1}^{n} d_j \tag{9}$$

⑥ Calculate the national happiness level based on the index weight and the product of each indicator.

## 4. Results and Discussion

### 4.1. The Verification Results of Real Estate Investment Growth on National Happiness

According to the empirical analysis, the data should be normalized before regression, and the existence of unit roots is tested in the panel data to avoid false regression. The research unveils the unit root test and the co-integration test on the explanatory variables and the explanatory variables. All the results rejected the null hypothesis, indicating no unit root and co-integration relationship in this panel data.

According to the five proxy variables, we used the entropy method to calculate the value of national happiness, which will not be elaborated here. From the definitions of the explanatory variables, *income* and *income_increase* were expected to be buoyant with national happiness. Thus, the actual coefficients between *realestate_increase* and *happiness* are original values. *gini*, *unemployment*, and *inflation* were expected to be negative with national happiness; the actual coefficients between *realestate_increase* and *happiness* were the opposite values.

The positive value of total effect indicates that the rapid growth of real estate investment has an overall positive effect on national happiness. The negative value indicates that the rapid growth of real estate investment has an overall negative effect on national happiness. This research uses EViews 9.0 to estimate the regression models. According to panel data features, the effectiveness of the mixed-effect model, the fixed-effect model, and the

random-effect model needs to be verified. With dependent variables *income*, *income_increase*, *gini*, *unemployment*, and *inflation*, control variables *pergdp_growth*, *educationcapital*, *healthcapital*, *tax*, *industry_structure*, and *socialsecurity*, and the explanatory variable *realestate_increase*, this research estimated the results of Equations (1)–(3) and the LR test. F statistics of cross-section, fixed period, and chi-squared values were significant under the 99% confidence level. However, when the cross-section and the time were both fixed, the adjusted $R^2$ took the maximum value. Therefore, the model excluded the mixed effect model and adopted the cross-section and fixed-time, fixed-effect model. Furthermore, the econometric Equation (1) regression equation was calculated using the random-effects model and was subjected to the Housman test, $p = 0.0000$. The null hypothesis was rejected. The same was used to prove that the fixed effect model should be used. Therefore, this research finally adopted double fixed effects of period and cross-section models (Tables 1 and 2).

**Table 1.** LS test of real estate investment growth on national happiness.

| | **Model 1** | **Model 2** | **Model 3** | **Model 4** | **Model 5** | **Model 6** |
|---|---|---|---|---|---|---|
| Explanatory Variables | *income* | *income_increase* | *gini* | *unemployment* | *inflation* | *happiness* |
| Method | | | LS-Least Square (and AR) | | | |
| *realestate_increase* | 0.0019 ** | 0.0068 ** | 0.0033 *** | 0.0003 * | 0.0005 ** | 0.0112 ** |
| *pergdp_growth$_{-1}$* | 0.0521 | −0.0071 | 0.0377 ** | 0.001738 | 0.0068 * | 0.0086 *** |
| *educationcapital$_{-1}$* | 0.2385 | 0.0042 | −0.1476 *** | 0.0442 *** | 0.0170 *** | 0.2874 ** |
| *healthcapital$_{-1}$* | −4.2813 *** | 3.4967 (0.2553) | −6.6031 ** | −0.6566 * | 0.1454 | 0.2154 * |
| *industry_structure$_{-1}$* | −0.9075 ** | 0.0964 | −0.1148 * | −0.0169 * | −0.0001 | 0.0547 ** |
| *tax$_{-1}$* | 5.6369 *** | 0.2691 | −0.3172 ** | −0.1371 *** | 0.0115 * | −0.2147 ** |
| *social_security$_{-1}$* | −3.3606 *** | 0.0407 | 0.0264 | 0.0075 | −0.0569 *** | 0.2474 *** |
| *Constant* | 1.9437 *** | −0.0409 | 0.5583 *** | 0.0596 *** | −0.0247 | 2.4791 * |
| Cross-section effect | Controlled | Controlled | Controlled | Controlled | Controlled | Controlled |
| Period effect | Controlled | Controlled | Controlled | Controlled | Controlled | Controlled |
| Adj. $R^2$ | 0.9586 | 0.6709 | 0.6671 | 0.7965 | 0.7374 | 0.6477 |
| F statistics | 211.8226 | 34.3805 | 26.1181 | 36.5995 | 40.1553 | 28.1246 |
| Observations | 558 | 558 | 558 | 558 | 558 | 558 |

Note: *, **, *** represent statistically significant levels of 10%, 5%, and 1%, respectively; the subscript -1 indicates that the variable is lagging by one period, and the following is the same.

Table 1 reports the test results of real estate investment growth on national happiness in various provinces in China. The LS method's regression results in column 1 show that under controlling a series of influencing factors, the growth of real estate investment increases the per capita disposable income of residents with a 95% confidence level. The second column shows the rapid growth of real estate investment and development in each province of China. The growth effect of resident disposable income is significantly positive at the 95% confidence level. The third column reported that the rapid growth of real estate investment and the effect of the Gini coefficient of disposable income in all provinces are negatively related at a 99% confidence level. The fourth column shows that the relationship between the growth rate of real estate investment and development and the unemployment rate is significantly positive at the 90% confidence level. The fifth column reports the significant positive effects at 95% confidence between the growth rate of real estate investment and inflation. The results of these five estimates show that the growth of real estate investment has a significant impact on national happiness in several dimensions,

calculated as 0.0112, indicating that the rapid growth of China's real estate has a significant positive impact on national happiness, regardless of the instrumental variables to the effect.

**Table 2.** TSLS test results of the impact of real estate investment growth on national happiness.

| | **Model 1** | **Model 2** | **Model 3** | **Model 4** | **Model 5** | **Model 6** |
|---|---|---|---|---|---|---|
| Explanatory variables | *income* | *income_increase* | *gini* | *unemployment* | *inflation* | *happiness* |
| Method | TSLS—Two-Stage Least Squares (and AR) | | | | | |
| *realestate_increase* | −0.2049 *** | −0.0331 ** | −0.0161 *** | −0.0049 * | 0.0017 ** | −0.2153 ** |
| *pergdp_growth$_{-1}$* | 0.1349 | 0.0089 ** | 0.01464 | −0.0001 | −0.0083 * | 0.0067 *** |
| *educationcapital$_{-1}$* | 0.2039 | −0.0025 *** | −0.0745 * | 0.0442 *** | 0.0261 *** | 0.3412 ** |
| *healthcapital$_{-1}$* | −4.4404 *** | 3.1895 ** | −3.8594 ** | −0.6216 * | 0.1270 | 0.1024 * |
| *industry_structure$_{-1}$* | −1.4310 * | −0.0047 * | −0.1764 ** | −0.0053 * | 0.0287 | 0.0321 ** |
| *tax$_{-1}$* | 5.0094 *** | 0.1479 *** | −0.1111 | −0.1233 *** | 0.0281 * | −0.5126 * |
| *social_security$_{-1}$* | −3.3237 *** | 0.0478 | 0.1996 | 0.0067 | 0.0289 *** | 0.4574 *** |
| *Constant* | −3.3237 *** | 0.0678 *** | 0.5719 *** | 0.0472 *** | −0.0284 | 1.2541 * |
| *Cross-section effect* | Controlled | Controlled | Controlled | Controlled | Controlled | Controlled |
| *Period effect* | Controlled | Controlled | Controlled | Controlled | Controlled | Controlled |
| Adj. $R^2$ | 0.9610 | 0.7205 | 0.8248 | 0.8244 | 0.8103 | 0.7147 |
| DWH Chi$^2$/F | 58.2314 | 49.2361 | 36.2193 | 42.3612 | 45.3216 | 37.4712 |
| F statistics | 40.1406 | 44.316 | 26.1181 | 36.5995 | 40.1406 | 34.6974 |

Note: *, **, *** represent statistically significant levels of 10%, 5%, and 1%, respectively.

Table 2 shows the regression results of the instrument variables we set above and the TSLS estimation method. Columns 1 and 2 show that under the premise of controlling a series of related influencing factors and adopting the instrumental variables to eliminate endogenous problems caused by two-way causation, the effects of the growth of real estate investment in China on disposable income and the growth of disposable income are 99% and 95%, respectively. Additionally, the absolute value of the regression coefficient increased, which shows that the investment in real estate has a significant inhibitory effect on the increase in residents' disposable income in China. The third column shows that the relationship between the growth rate of real estate development investment and the Gini coefficient is significantly negative at the 95% confidence level.

Moreover, the regression coefficient's absolute value is more extensive, indicating that the growth of investment in real estate development is conducive to narrowing the income gap among residents. Column 4 reports that the growth of real estate investment and development and the unemployment rate is significantly negative at 10% of the statistical level. The growth rate of real estate investment increases by 1%, the unemployment rate of China decreases by 0.0049% correspondingly, and the real estate sector makes a positive contribution to employment promotion. The fifth column shows that the relationship between real estate investment growth and the inflation rate is more significant than LS model analysis results, indicating that the rapid growth of real estate investment has raised inflation in China. From the practical analysis of the instrumental variables set in this research, the DWH test results of the variables in Table 2 reject the null hypothesis that there is no endogenous problem at 1% statistical level; thus, we can confirm that the original equation has endogenous problems.

On the other hand, when taking TSLS two-stage instrumental variables, the real estate investment growth rate and the coefficients' absolute value significantly increased. The *p*-value became smaller than before, and the adjusted $R^2$ increased. In the first stage, the estimated F value was greater than 10. Therefore, the 16.38 cut-off value at a 10% error level

verified that instrumental variables' selection is appropriate [29,30]. When introducing the instrument variables, we found that the coefficient of national happiness and real estate investment decreased to $-0.2153$. The above test results from different angles for research Hypothesis 1 provide evidence of support. The real estate sector in China under rapid growth has a significant inhibitory effect on residents who have provided empirical evidence.

*4.2. Test Results of the Mechanism of Loan Preference*

This section will further explore how the rapid growth of real estate investment in China further hinders national happiness in China through the transmission mechanism of loan structures in various provinces and regions. Based on the econometric Equation (1), this research introduces the transmission mechanism of loan structure, setting the model as follows:

$$happiness_{it} = \beta_0 + \beta_1 realestate\_increase_{it} + \beta_2 loadstrucure \times realestate\_increase_{it} + \delta Z + \varepsilon_{it} \quad (10)$$

Compared with Equation (1), econometric Equation (10) introduces the interaction variable *loanstructure* × *realestate_increase* between real estate investment growth rate and loan structure to observe how the loan structure affects national happiness growth real estate investment. The *loanstructure* is the ratio of the year-end balance of small-, medium-, and micro-sized enterprises in each province's banking sector and the loan balance of real estate enterprises as of the end of the year. The smaller the value, the more bank-oriented financial institutions tend to provide loans to the real estate sector. The above financial institution's loan data were derived from the China Real Estate Financial Report.

Consider introducing an instrumental variable into Equation (10). According to the relationship between the instrumental and endogenous variables, when the number of instrumental variables is greater than or equal to the endogenous variable, the equation is solvable. There are two endogenous variables in Equation (10), *realestate_increase*, *loanstructure* × *realestate_increase*, and one in Equation (1), *construction_land*. Thus, it needs at least one instrumental variable. This research selects the financial institutions' loan term structure in Equation (10), representing the ratio of the total values of short-term loan and medium and long-term loans at the bank system, identified as *loan_structure*. There are two reasons for the selection. First, a short-term bank loan is mainly designed for individuals and microenterprises, while the customers of medium and long-term bank loans focus on large enterprises with great capital turnover demands, such as the real estate sector. The *loan_structure* and *loanstructure* are related closely, accorded with the high correlation principle for the instrumental variable. Second, there is no direct correlation between the loan's term structure and the control variables in the financial institutions. Therefore, it is reasonable to use *loan_structure*.

Table 3 reports the regression results of Equation (10). Specifically, the first column estimates show that the coefficient of real estate investment growth rate in each province in China is significantly positive at the 5% statistical level. The interaction items with the financial institutions' loan structure are significantly negative at the 10% statistical level. From the connotation analysis of variables, this conclusion is theoretically reasonable. Tables 1 and 3 show that the absolute value of each variable's estimated coefficient in the econometric Equation (10) introduced into the loan structure has significantly increased. It proves that financial institutions' loan preference plays an essential role in the growth of real estate investment and the role of a booster between real estate investment growth and national happiness. Financial institutions have aggravated the real estate sector's impact on national happiness by adjusting the loan structure.

The same findings are confirmed in the TSLS estimates in Table 4. Table 4 shows the results of the TSLS that introduced the new instrument variable *loan_structure*. The corresponding coefficient of *happiness* is $-0.3099$. Compared with Table 2, the absolute coefficients of national happiness have significantly increased when introducing loan preference. It provides empirical evidence for Hypothesis 2 that the growth rate of investment

in real estate development in various provinces and regions in China has a significant inhibitory effect on promoting happiness among residents.

**Table 3.** LS test results of real estate investment growth on national happiness through loan structure.

| | Model 1 | Model 2 | Model 3 | Model 4 | Model 5 | Model 6 |
|---|---|---|---|---|---|---|
| Explanatory variables | *income* | *income_increase* | *gini* | *unemployment* | *inflation* | *happiness* |
| Method | | | LS-Least Square (and AR) | | | |
| *realestate_increase* | 0.0383 ** | 0.0072 ** | 0.0043 *** | −0.0009 * | 0.0028 ** | 0.1445 * |
| *R* loanstructure* | −0.037 * | −0.0004 * | 0.0013 ** | 0.0013 ** | −0.0025 ** | 0.0102 ** |
| *pergdp_growth-1* | 0.047 | −0.0072 ** | 0.0068 ** | 0.0019 | −0.0081 * | 0.2914 * |
| *educationcapital-1* | 0.2122 | 0.0039 *** | −0.0562 *** | 0.0451 *** | 0.0241 *** | 0.2179 * |
| *healthcapital-1* | −4.5232 *** | 3.4693 * | −2.2881 ** | −0.5755 * | −0.0475 | 0.3041 |
| *industry_structure-1* | −1.0182 ** | 0.0951 *** | −0.1399 ** | −0.0131 | 0.0177 | 0.0749 ** |
| *tax-1* | 5.5945 *** | 0.2686 | 0.0249 ** | −0.1357 *** | 0.0210 * | −0.2798 * |
| *social_security-1* | −3.3395 *** | 0.0410 *** | 0.2311 * | 0.0068 | 0.0306 *** | 0.2547 * |
| *Constant* | 2.0511 | −0.0395 | 0.514459 ** | 0.0557 *** | −0.0168 | 2.7489 * |
| Cross-section effect | Controlled | Controlled | Controlled | Controlled | Controlled | Controlled |
| Period effect | Controlled | Controlled | Controlled | Controlled | Controlled | Controlled |
| Adj.$R^2$ | 0.9588 | 0.7526 | 0.6787 | 0.7979 | 0.8116 | 0.7847 |
| F statistics | 208.7014 | 42.9596 | 19.8542 | 36.2267 | 39.4113 | 35.4726 |
| Observations | 558 | 558 | 558 | 558 | 558 | 558 |

Note: *, **, *** represent statistically significant levels of 10%, 5%, and 1%, respectively; *R* loanstructure* represents *realestate_increcrease* × loanstructure.

**Table 4.** TSLS test results of the impact of real estate investment through the loan preference on national happiness.

| | Model 1 | Model 2 | Model 3 | Model 4 | Model 5 | Model 6 |
|---|---|---|---|---|---|---|
| Explanatory variables | *income* | *income_increase* | *gini* | *unemployment* | *inflation* | *happiness* |
| Method | | | TSLS—Two-Stage Least Squares (and AR) | | | |
| *realestate_increase* | −0.2919 ** | −0.0494 ** | 0.0267 *** | −0.0072 ** | 0.0025 ** | −0.3099 ** |
| *R* loanstructure* | 0.2775 * | 0.0518 ** | 0.0223 ** | −0.0073 ** | −0.0027 ** | 0.7847 * |
| *pergdp_growth$_{-1}$* | 0.1004 | 0.002462 ** | 0.0113 ** | 0.0008 | −0.0080 * | 0.0072 ** |
| *educationcapital$_{-1}$* | 0.4303 | 0.0397 *** | −0.0420 *** | 0.0390 *** | 0.0240 *** | 0.4127 * |
| *healthcapital$_{-1}$* | −2.5013 *** | 6.8083 * | −0.9516 ** | −1.1335 * | −0.0599 | 0.1047 ** |
| *industry_structure$_{-1}$* | −0.1447 ** | 0.2353 *** | −0.0874 ** | −0.0393 | 0.0163 | 0.0358 * |
| *tax$_{-1}$* | 5.8798 *** | 0.310345 | 0.0369 ** | −0.1463 *** | 0.0197 * | −0.5417 ** |
| *social_security$_{-1}$* | −3.5131 *** | 0.01251 *** | 0.2199 * | 0.0117 | 0.0307 *** | 0.6574 * |
| *Constant* | 1.1352 ** | −0.1864 *** | 0.4594 *** | 0.0831 *** | −0.0153 | 1.4758 * |
| Cross-section effect | Controlled | Controlled | Controlled | Controlled | Controlled | Controlled |
| Period effect | Controlled | Controlled | Controlled | Controlled | Controlled | Controlled |
| Adj. $R^2$ | 0.9382 | 0.7621 | 0.6896 | 0.8064 | 0.8117 | 0.7984 |
| DWH Chi$^2$/F | 62.3125 | 52.2631 | 25.3616 | 53.2361 | 54.3261 | 57.2426 |
| F statistics | 208.4975 | 34.0998 | 19.9569 | 35.8566 | 39.2756 | 37.1792 |

Note: *, **, *** represent statistically significant levels of 10%, 5%, and 1%, respectively; *R* loanstructure* represents *realestate_increcrease* × loanstructure.

Moreover, through the channel of influence of the financial institutions' loan propensity, the inhibitory effect on residents' happiness is even more prominent. There is a typical financial repression system in China at this stage. Financial institutions such as monopolistic banks tend to provide loans to the real estate industry to meet the huge capital needs for the rapid expansion of the real estate industry. There is evidence that the financial loan market is typical of a buyer's market when faced with the real estate sector from 2000 to 2018 in China. The real estate sector has a very high voice. The banking sector often seizes the corporation opportunities through raising the loan amount, interest rates, and lower access. It further reduces the space for individuals and other enterprises to obtain loans, which is not conducive to sustainable economic development.

### 4.3. Retesting Results of Conduction Mechanism: Based on the Mediating Effect Test Model

The above regression model conducts a preliminary analysis of the transmission mechanism of the financial institutions' loan structure to national happiness based on the endogenous problem between real estate investment growth and national happiness. The interaction item's coefficients between the growth rate of real estate and the financial institutions' loan structure are significant at the statistical level. However, it is likely to reveal only the endogenous interactive relationship between the rapid expansion of the real estate sector and the loan preference of China's financial institutions t. The interactive relationship has possibly dampened the happiness of residents. The model may not be able to identify the rapid growth of the real estate sector in Hypothesis 2 effectively, which further inhibits national happiness through the transmission mechanism of the financial institutions' loan structure. The classic intermediary test method is the mediating effect test model developed by Baron, Kenny, Chengand Wen [31–34].

To verify the effect of loan preference on real estate investment and national happiness, we adopted the two-stage least squares method. The two-stage least squares method analyzes the interaction of hidden variables, and there is no restriction on the distribution of variables. More importantly, according to econometrics, when there is a two-way effect between the independent and dependent variables, the two-stage least squares method is required. We have discussed that real estate investment and national happiness have a two-way causal relationship. Therefore, choosing the two-stage least squares method can avoid discussing the distribution of variables and conform to the two-way causal relationship between real estate investment and happiness, which is a proper choice to test the conduction mechanism shown in Figure 1.

$$happiness_{it} = \beta_0 + \beta_1 realestate\_increase_{it-1} + \delta \cdot Z_{it-1} + \varepsilon_{it} \tag{11}$$

$$loadstructure_{it} = \alpha_0 + \alpha_1 realestate\_increase_{it-1} + \eta.Z_{t-1} + v_{it} \tag{12}$$

$$happiness_{it} = \kappa_0 + \kappa_1 realestate\_increase_{it-1} + \kappa_2 loadstructure_{it-1} + \theta.Z_{t-1} + \mu_{it} \tag{13}$$

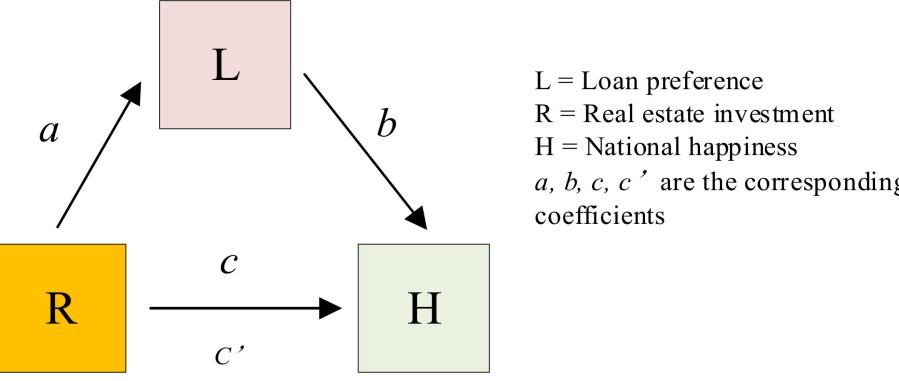

**Figure 1.** Real estate growth, loan preference, and national happiness.

The first step is to regress the econometric model (Equation (11)) and test whether the regression coefficient of one period lagged real estate investment growth is significantly negative. If so, it means that the rapid growth of investment in the real estate sector hurts national happiness in China. The second step tests whether the coefficient of the rapid growth of investment in the real estate sector and that of the financial institutions of the intermediary variable $\alpha_1$ is significantly negative in Equation (12). If significantly negative, it indicates that real estate investment's rapid growth harms the financial institutions' loan structure. The third step is to judge the status between $k_1$ and $k_2$. If $k_1$ is negative significantly, $k_2$ is significantly positive, and $k_1$ is smaller than $\beta_1$, which proves the existence of a partial mediation effect. If $k_1$ is not significant and $k_2$ is significant, it proves a full mediation effect of China's financial institutions' loan structure.

Table 5 reports our regression results using the above recursive model to test the effect of the rapid growth of real estate investment in China through the transmission mechanism of the financial institutions' loan preference on the impact of regional and national happiness. The regression results of models 1–6 in Table 6 show that the rapid growth of real estate investment is negatively correlated with disposable income, income growth, and unemployment rate at or more than the 90% statistical level, except for Gini coefficient and inflation, while the total effect coefficient of real estate investment growth in model 6 is −0.1723. The regression results in Table 6 show that the rapid growth of real estate investment in China has a significant inhibitory effect on residents' happiness in the region. The rapid growth of investment in the real estate sector has a significant impact on the financial institutions' loan preference.

**Table 5.** Intermediary effect test results of real estate investment impact national happiness through loan preference (Steps 1 and 2).

| Reg. Steps | The First Step | | | | | | The Second Step |
|---|---|---|---|---|---|---|---|
| Model | Model 1 | Model 2 | Model 3 | Model 4 | Model 5 | Model 6 | Model 7 |
| Explanatory variables | *income* | *income_increase* | *gini* | *unemployment* | *inflation* | *happiness* | *loanstructure* |
| *tealestate_increase_$_{-1}$* | −0.1989 ** | −0.0322 ** | 0.0192 *** | −0.0047 * | 0.0016 ** | −0.1723 ** | −0.1228 *** |
| *pergdp_growth_$_{-1}$* | 0.1408 | 0.0098 ** | 0.0148 ** | −0.0002 | −0.0084 * | 0.0125 * | 0.0852 * |
| *educationcapital_$_{-1}$* | 0.2038 | −0.0025 *** | −0.0602 *** | 0.0449 ** | 0.0261 *** | 0.4257 ** | 0.0502 |
| *healthcapital_$_{-1}$* | −4.4251 *** | 3.2141 * | −2.4918 ** | −0.6252 * | 0.1258 | 0.1247 * | 2.1975 ** |
| *Industry_structure_$_{-1}$* | −1.4192 ** | −0.0028 *** | −0.1898 ** | −0.0056 | 0.0286 | 0.0217 * | −0.4042 |
| *tax_$_{-1}$* | 5.0342 *** | 0.1519 | −0.0308 ** | −0.1239 *** | 0.0279 * | −0.5847 ** | −0.7394 |
| *social_security_$_{-1}$* | −3.3305 *** | 0.0468 *** | 0.2345 * | −0.1239 | 0.0289 *** | 0.6747 * | 0.3737 * |
| *Constant* | 2.4828 ** | 0.0654 ** | 0.5677 *** | 0.0069 *** | −0.0283 | 1.7421 * | 0.2694 |
| Cross-section effect | Controlled | Controlled | Controlled | Controlled | Controlled | Controlled | Controlled |
| Period effect | Controlled | Controlled | Controlled | Controlled | Controlled | Controlled | Controlled |

Note: *, **, *** represent statistically significant levels of 10%, 5%, and 1%, respectively.

Table 6 shows the regression results of the third step of the mediated effect model's test results. It shows that the absolute coefficient of real estate investment growth in all regions is greater in the third step than the value in the first step. It verifies that the loan preference plays a part in the intermediary effect, which shows that the rapid growth of real estate investment can further inhibit national happiness in all regions through China's loan structure.

**Table 6.** Intermediary effect test results of real estate investment impact national happiness through the financial institutions' loan structure (third step).

| Reg. Steps | The Third Step | | | | | |
|---|---|---|---|---|---|---|
| Models | Model 1 | Model 2 | Model 3 | Model 4 | Model 5 | Model 6 |
| Explanatory variables | *income* | *income_increase* | *gini* | *unemployment* | *inflation* | *happiness* |
| *realestate_increase$_{-1}$* | −0.2112 ** | −0.0347 ** | 0.0242 *** | −0.0061 ** | 0.0018 ** | −0.2246 ** |
| *loanstructure$_{-1}$* | 0.2336 *** | 0.0438 ** | 0.0186 ** | −0.0061 ** | −0.0023 ** | 0.0217 * |
| *pergdp_growth$_{-1}$* | 0.1351 | 0.0088 ** | 0.0136 ** | −0.0002 | −0.0082 * | 0.3147 * |
| *educationcapital$_{-1}$* | 0.1956 | −0.0031 *** | −0.0562 *** | 0.0623 ** | 0.0461 *** | 0.2174 |
| *healthcapital$_{-1}$* | −2.5621 *** | 3.2561 * | −2.4618 ** | −0.9352 * | 0.3258 | 0.0274 * |
| *industry_structure$_{-1}$* | −1.7162 ** | −0.0038 *** | −0.2011 ** | −0.0056 | 0.0282 | −0.2147 |
| *tax$_{-1}$* | 5.1265 *** | 0.1719 | −0.0356 ** | −0.1241 *** | 0.0282 * | −0.6641 * |
| *social_security$_{-1}$* | −3.3805 *** | 0.0612 *** | 0.2315 * | −0.1939 | 0.0279 *** | 0.4170 * |
| *Constant* | 3.4128 ** | 0.0514 ** | 0.5637 *** | 0.0089 *** | −0.0263 | 0.9848 * |
| Cross-section effect | Controlled | Controlled | Controlled | Controlled | Controlled | Controlled |
| Period effect | Controlled | Controlled | Controlled | Controlled | Controlled | Controlled |

Note: *, **, *** represent statistically significant levels of 10%, 5%, and 1%, respectively.

## 5. Conclusions and Policy Implications

The real estate market in China is under the combined effect of China's specific urbanization background, the dependence of local governments on land finance, the speculative demand of the real estate market caused by the lack of proper investment channels, and the financial domain dominated by monopolistic banking institutions. Thus, the real estate market exhibits the critical features of rapid expansion and even the real estate bubble. In the context of China's 13th and 14th Five-Year Plans, it is natural for people to consider whether China's massive scale of real estate investment and its rapid growth rate accompanied by high housing prices will negatively affect national happiness. Moreover, there is a typical system of financial repression in China. The loan priority of financial institutions represented by monopolistic banks to the real estate sector will further inhibit national happiness.

To answer these questions, we used panel data from 2000 to 2018 at the provincial level in China, using the new construction land per capita and the financial institutions' loan structure as the instrument variables, and made some meaningful discoveries. Firstly, the rapid growth of real estate investment has had a significant adverse effect on China's national happiness. Although real estate investment has positive effects on employment and income growth, the total effect is significantly harmful. Specifically, rapid real estate investment is conducive to narrowing the income gap and increasing employment. However, it harms the residents' disposable income and even exacerbates inflation, undermining national happiness.

Secondly, the rapid growth of real estate investment has further dampened national happiness through financial institutions' loan preference. In the econometric model of introducing financial institutions' loan structure, the concrete manifestation is that the absolute value of the elasticity coefficient of real estate investment growth rate and national happiness is significantly more generous. The total effect is significantly enhanced. The followed intermediary effects model confirmed the transmission mechanism from the rapid growth in real estate investment through loan structure to national happiness.

These findings provide reference values for maintaining a healthy and stable real estate market and raising residents' well-being during urbanization in China. The government should attach great importance to the unlimited growth of real estate investment and

harm the sustainable development of China's economy and national happiness. To raise national happiness, the government may need to properly understand the residential demand of real estate, curb speculative investment in the real estate market, and adjust local governments and the real estate sector's interests. Thus, it will eliminate local bubbles in real estate, correct the misallocation of resources caused by the real estate sector's irrational development, and ease the adverse effects of the real estate sector's rapid growth on national happiness. Additionally, the government should exert macro-control on the loan preferences of financial institutions, utilizing finance and tax to reduce the inhibiting effects of financial loan preference on national happiness.

There is no precise measurement standard for the definition of happiness in academia; therefore, this study mainly selected indicators that were easy to quantify to measure happiness due to scientific considerations: objective happiness. However, the subjective feeling is still a vital measure of happiness, although it is not included in this research, which is the limitation of this article. Therefore, the combination of objective well-being and subjective well-being is an important research direction in the future.

**Author Contributions:** Conceptualization: S.K. and X.L.; methodology: S.K.; software: S.K.; funding acquisition: S.K. Both authors have read and agreed to the published version of the manuscript.

**Funding:** This research was funded by self-determined research funds of CCNU from the colleges' basic research and operation of MOE, grant number CCNU20A03005, Chinese National Natural Science Fund, grant number 72064042, and Philosophy and Social Science Research Project in Yunnan Province, grant number QN202026 and the APC was funded by high-end talent introduction plan of Central China Normal University.

**Data Availability Statement:** All the data were from the China Statistical Yearbook and the China Statistical Yearbook 2000–2019 online at http://www.stats.gov.cn/tjsj/ndsj/.

**Acknowledgments:** The paper is financially supported by self-determined research funds of CCNU from the colleges' basic research and operation of MOE (CCNU20A03005), high-end talent introduction plan of Central China Normal University, Chinese National Natural Science Fund (72064042), and Philosophy and Social Science Research project in Yunnan Province (QN202026). In writing this paper, the authors received much care and guidance from the authors' friends and colleagues. Special thanks are given to the anonymous reviewers for their constructive suggestions.

**Conflicts of Interest:** The authors declare no conflict of interest.

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
