# Peer review of "Real Estate Investment, Loan Preference, and National Happiness: Evidence from China"

_land, doi:10.3390/land10040428_

Round 1

Reviewer 1 Report

This is a very good paper on real estate investment, loan preference, and national happiness in China. Before it is publishable, authors should address several minor issues:

1)The introduction should better position this study on the happiness and state investment literature by highlighting what it brings new in the existing literature;

2) The literature review could be a bit enlarged by mentioning several other studies on happiness, see the special issue on happiness studies entitled 'The life of a happy worker: Examining short-term fluctuations in employee happiness and well-being' in Human Relations journal, 2010. Also, some unhappiness issues could be shortly mentioned around the world, in order to state that sometimes real estate investments could lead not to happiness but to stigma and unhappiness (authors can give 1-2 examples in the world, see for instance how IKV, a state investing estate company in Hungary built new residents for marginal people and this lead to overmarginalisation and unhappiness - doi: 10.5937/gp24-28226).

3) Conclusions should shortly include limitations of the study and follow-up esearch (eg. how other researchers can develop further certain findings in this paper).

Author Response

Comment #1:This is a very good paper on real estate investment, loan preference, and national happiness in China. Before it is publishable, authors should address several minor issues:

1)The introduction should better position this study on the happiness and state investment literature by highlighting what it brings new in the existing literature;

Response #1: We have added the content about the contribution in reality and theory in the introduction part. Here is the added content:

The main contributions of this paper are as follows. Although existing research has focused on the relationship between housing and residents' happiness, more on the social and economic value of housing itself, few scholars have paid attention to the impact of continued growth in real estate investment on residents' happiness. In China's economic and cultural environment, large-scale real estate investment and speculation have caused housing prices to remain high, and the symbolic significance of wealth and status attached to real estate has been highly magnified. Its impact on Chinese residents' happiness is undeniable. Studying the mechanism of China's real estate investment's influence on residents' happiness can enhance theoretical research on real estate investment and happiness. It can also help the government better cope with real estate investment's negative impact on residents' happiness.

The existing literature does not pay attention to the critical influence of loan preference on the relationship between real estate investment and residents' happiness. This research can help us understand how loan preference affects the relationship between the two factors and deepen the understanding of loan preference's influence on the "real estate investment-resident happiness" framework. The intermediary effect of "sense" reveals the mechanism and influence of intermediary effect, and at the same time, can provide policy reference for decision-making departments.

Comment #2:The literature review could be a bit enlarged by mentioning several other studies on happiness, see the special issue on happiness studies entitled 'The life of a happy worker: Examining short-term fluctuations in employee happiness and well-being' in Human Relations journal, 2010. Also, some unhappiness issues could be shortly mentioned around the world, in order to state that sometimes real estate investments could lead not to happiness but to stigma and unhappiness (authors can give 1-2 examples in the world, see for instance how IKV, a state investing estate company in Hungary built new residents for marginal people and this lead to overmarginalisation and unhappiness - doi: 10.5937/gp24-28226).

Response #2: According to the suggestion, we have added some literature, including the reviewer's suggested papers.

Comment #3: Conclusions should shortly include limitations of the study and follow-up research (eg. how other researchers can develop further certain findings in this paper).

Response #3: We have added the limitations and follow-up research in the end of the paper. Here is the added content:

   Because there is no precise measurement standard for the definition of happiness in academia, this article mainly selects indicators that are easy to quantify to measure happiness due to scientific considerations, objective happiness. However, the subjective feeling is still a vital content of measuring happiness, but it is not included in this research, which is the limitation of this article. Therefore, the combination of objective well-being and subjective well-being is an important research direction in the future.

Reviewer 2 Report

The authors tried to verify the existence of an inverse relationship between growth in real estate investments and national happiness in China. This is not a direct one, but one that passes through the inhibitory effect of the loan preference of financial organizations. In order to verify if the relationship between the two variables is not spurious, the authors use Two-stage Least Squares and LS models. The results are supported by a series of indicators that give validity to the hypotheses, however I would like to suggest that perhaps the most suitable approach to challenge this type of phenomenon is the structural equation modeling. I therefore invite the authors to justify the choice of the model used.

Author Response

Reviewer #2

Comment #1:The authors tried to verify the existence of an inverse relationship between growth in real estate investments and national happiness in China. This is not a direct one, but one that passes through the inhibitory effect of the loan preference of financial organizations. In order to verify if the relationship between the two variables is not spurious, the authors use Two-stage Least Squares and LS models. The results are supported by a series of indicators that give validity to the hypotheses, however I would like to suggest that perhaps the most suitable approach to challenge this type of phenomenon is the structural equation modeling. I therefore invite the authors to justify the choice of the model used.

Response #1:We have to accept that the structural equation model is a feasible solution to this problem, but the two-stage least squares method also has its advantages. The two-stage least squares method analyzes the interaction of hidden variables, and there is no restriction on the distribution of variables. The variable can be used whether it is a normal distribution or a non-normal distribution. More importantly, according to the econometrics tutorial, when there is a two-way effect between the independent and dependent variables, the two-stage least squares method is required. In the paper, we have pointed out in paper 3.2 when discussing instrumental variables that the real estate investment is Well-being has a two-way causal relationship, and the two influence each other. Therefore, choosing the two-stage least squares method can avoid discussing the distribution of variables and conform to the two-way causal relationship between real estate investment and happiness, which is a proper choice.

Reviewer 3 Report

The paper is suitable for publication.

Author Response

Thanks.
